# Azobenzene-containing liquid crystalline composites for robust ultraviolet detectors based on conversion of illuminance-mechanical stress-electric signals

Xiaoxiong Zheng[1,2], Yining Jia[1,2] & Aihua Chen [1,2✉]

Wearable ultraviolet (UV) detectors have attracted considerable interest in the military and civilian realms. However, semiconductor-based UV detectors are easily interfered by elongation due to the elastic modulus incompatibility between rigid semiconductors and polymer matrix. Polymer detectors containing UV responsive moieties seriously suffer from slow response time. Herein, a UV illuminance–mechanical stress–electric signal conversion has been proposed based on well-defined ionic liquid (IL)-containing liquid crystalline polymer (ILCP) and highly elastic polyurethane (TPU) composite fabrics, to achieve a robust UV monitoring and shielding device with a fast response time of 5 s. Due to the electrostatic interactions and hydrogen bonds between ILs and LC networks, the ILCP-based device can effectively prevent the exudation of ILs and maintain stable performance upon stretching, bending, washing and 1000 testing cycles upon 365 nm UV irradiation. This work provides a generalizable approach toward the development of full polymer-based wearable electronics and soft robots.

[1] School of Materials Science and Engineering, Beihang University, Beijing, People's Republic of China. [2] Beijing Advanced Innovation Centre for Biomedical Engineering, Beihang University, Beijing, People's Republic of China. ✉email: chenaihua@buaa.edu.cn

For the past decade, wearable electronics have been rapidly developed for envisioned applications in a wide range of areas such as healthcare devices[1], environmental monitors[2], power supplies[3], and Internet of Things technologies[4], because of their multiple sensing capabilities, unique flexibility, and portability[5]. Among the versatile wearable electronics, ultraviolet (UV) detectors have garnered extensive attention in recent years for their significant applications in the military and civilian realms, such as defense warning systems[6], photochemical synthesis[7], medical treatments[8], security communication[9], etc. However, due to the strong penetrating ability, excess UV exposure can reach deep skin layers, leading to serious health hazards such as skin ageing and skin cancer[10]. In this regard, it is highly desired to achieve a wearable all-in-one UV monitoring and shielding device for daily life and industrial production under UV irradiation[11]. Previous works have reported wearable UV detectors by geometrically patterning rigid photoelectric semiconductors (such as zinc oxide) on a flexible polymer matrix, featuring superior sensitivity[12,13]. However, the devices are prone to occur delamination under elongation because of the elastic modulus incompatibility between rigid semiconductors and flexible matrix, resulting in a serious decrease in performance. In addition, due to the poor conformability and processing capability of rigid semiconductors, more suitable UV shielding materials are desired for wearable devices.

Based on polymer chemistry, functional polymer materials have realized the integration of skin-inspired properties in electronics, such as stretchability, conformability, adhesion, self-healing and degradability[14,15]. To fabricate polymer-based UV detectors, photoisomerizable moieties (e.g., azobenzene, spiropyran and diarylethene) are generally introduced into polymers[16]. Under UV irradiation, the photoisomerizable moieties will induce physical and chemical property changes of polymers, such as color or polarity. To date, reported flexible polymer-based UV detectors mainly exhibit visual color changes without electric signal output, which hinders the integration and application in sensor systems[17,18]. The UV induced physical and chemical property changes can be converted to electric (capacitance or current) signals via incorporating conductive fillers into the polymer matrix, such as carbon nanotubes, silver nanowires and ionic liquids (ILs)[19,20]. Among these, ILs are attracting growing attention because of their high ionic conductivity, chemical and thermal stability, nonvolatility, and low flammability[21,22]. For instance, Watanabe and coworkers reported a reversible ion-conducting switch by mixing azobenzene molecules into poly(N-isopropyl acrylamide) (PNIPAm) IL gels[19]. Due to the *trans*-to-*cis* isomerization under UV light, the increased polarity of azobenzene triggers the macroscopic sol-gel transition of IL gels, resulting in the increase of ionic conductivity. However, physically crosslinked PNIPAm IL gels suffer from poor mechanical properties, exudation of ILs, and a slow response time of 6 min. Alaniz and coworkers designed poly(ethylene oxide) films covalently grafted diarylethene-containing ILs, using for the light-modulation of ionic conductivity[20]. Diarylethene rings reversibly switch between open-close state under alternative UV/visible light, leading to the charge differences and thus the change of ionic conductivity. Nevertheless, the films exhibited a long response time of 15 min. Till now, the reported works mainly concentrated on the polarity modulation of photoisomerizable moieties via illumination to control the ionic conductivity. Generally, it will take some time for photoisomerizable moiety-containing polymers to reach the photostationary state upon illumination[19,20,23]. The response time is substantially decided by the polymer selection and composite structures. Therefore, an elaborate polymer composite of UV responsive polymers effectively integrated with ILs is highly desired for sensitive UV detectors.

Liquid crystalline elastomers (LCEs), or crosslinked liquid crystalline polymers (CLCPs), are highly sensitive to environmental stimuli owing to the synergistic effect of liquid crystalline (LC) alignment and polymer networks[24,25]. Azobenzene moieties are always functionalized into LC networks to achieve light-driven deformation of CLCPs. Based on the *trans-cis* isomerization of azobenzene mesogens under alternative UV and visible light or heating, the azobenzene-containing (Azo) CLCPs undergo a reversible phase transition between an LC phase and an isotropic state, generating internal stress and leading to reversible shape changes[26]. It is reported that based on reasonably designed structures, as long as 1 mol% of azobenzene mesogens reach to the photostationary state upon illumination, the generated energy can lead to the phase transition of the whole systems, featuring a fast response time of several seconds for Azo-CLCPs[27,28]. If integrated with ILs by feasible structural designs, the stress induced by phase transition has the potential to directly facilitate the ion migration of ILs, and rapidly transform to current or resistance signals. Thus, CLCPs can be utilized as wearable electronics, benefiting from the rapid response ability and robust mechanical properties. However, CLCPs are conventionally fabricated by one-pot copolymerization of LC monomers and crosslinkers in LC cells[24]. Thus, CLCPs generally exhibit poor processability because of the insoluble and infusible crosslinked networks. To the best of our knowledge, CLCPs used as wearable UV detectors or other flexible electronics have not been reported yet. We previously reported a solution-processable CLCPs consisting of azobenzene-containing block copolymers (Azo-BCPs) and branched polyethyleneimine (PEI), as shown in Fig. 1a[29]. CLCPs can be fabricated by a two-step post-crosslinking method, which involves the molding of non-crosslinked LCPs and the subsequent crosslinking in PEI solution.

Herein, we demonstrate IL-containing liquid crystalline polymers (ILCPs) via introducing hydrophobic ILs into the aforementioned Azo-CLCP matrix for intrinsically flexible and highly sensitive UV detectors. To improve the mechanical property, thermoplastic polyurethane (TPU) is selected to mix with Azo-BCPs to fabricate LCP-TPU fabrics through electrospinning (Fig. 1b), benefiting from their solution-processabilities. Then, the CLCP-TPU fabrics are obtained by the post-crosslinking of LCP-TPU fabrics in PEI solution, capable of portability and air permeability[30]. The high porosity and large surface area of fabrics also accelerate the penetration of ILs for preparing ILCP fabrics. Benefiting from the cooperative effects of Azo-LC alignment and polymer networks, the ILCP-based device constructs a rapid conversion of UV illuminance–mechanical stress–electric signals, which substantially facilitates the sensitivity of UV detectors (Fig. 1c). In addition, the abundant amino groups in PEI and urethane groups in TPU make the device easily adhere to skins by multiple hydrogen bonds (Supplementary Movie 1). Moreover, the exudation of ILs is effectively suppressed by the strong electrostatic interactions and hydrogen bonds between amino groups and ions, as shown in the FTIR spectra (Supplementary Fig. 2). Thus, the ILCP fabrics can be utilized as a flexible, stretchable, and washable UV monitoring and shielding materials. The ILCP-based device demonstrates an illuminance detection range of 10 ~ 270 mW cm$^{-2}$, stability of 1000 testing cycles and especially a response time of 5 s under 365 nm UV light. Owing to the electric signal output, we have designed the wearable ILCP-based on-demand information encoding electronics for UV security unlock and input via Internet of Things technologies.

## Results

**Fabrication of ILCP fabrics**. In the fabrication process, firstly, the LCP-TPU fabrics were prepared by electrospinning mixture

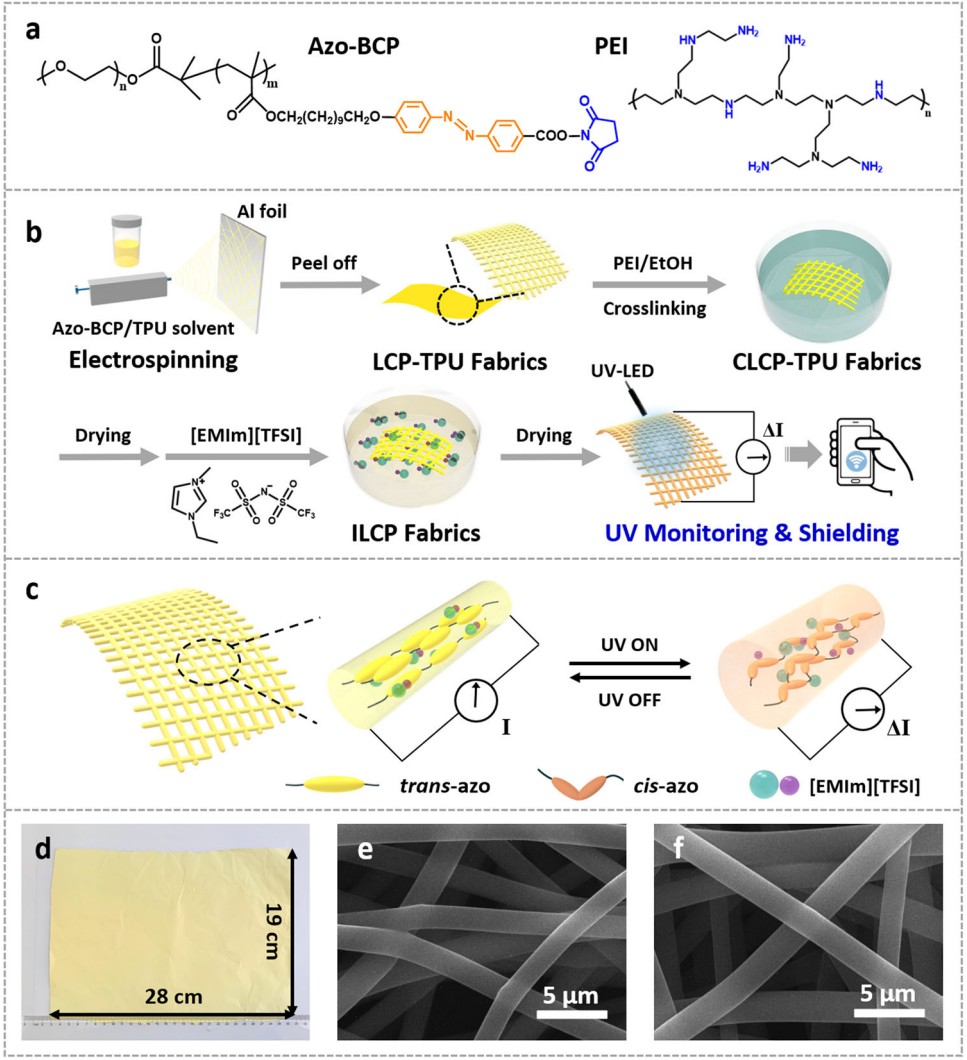

**Fig. 1 Fabrication of ILCP fabrics. a** Chemical structures of the Azo-BCP and PEI used in this study. **b** Experimental schematic illustration of the fabrication process of ILCP fabrics. **c** Schematic illustration of the structure and mechanism for the ILCP-based device. **d** The photograph of as-fabricated large-scale LCP-TPU fabrics with a size of 28 cm × 19 cm × 20 μm. SEM images of **e** CLCP-TPU fibers and **f** ICLP fibers.

solution of Azo-BCPs and TPU onto an aluminum foil collector in large scale (Fig. 1d). To obtain homogeneous solutions, the Azo-BCPs and TPU were sufficiently dissolved in DMF/THF solvents via stirring for 12 h due to their high molecular weights. The Azo-BCP/TPU system demonstrates enough compatibility because of the 11 wt.% hydrophilic PEO blocks in Azo-BCPs, as shown in the dynamic mechanical analysis (Supplementary Fig. 3a). From UV-Vis absorption spectra (Supplementary Fig. 3b), the Azo-BCP/TPU fabrics exhibits 365 nm UV absorption in each weight ratios, ensuring the UV-sensing ability. Secondly, the obtained LCP-TPU fabrics were immersed in an ethanol solution of PEI for 6 h to complete the post-crosslinking reaction to form CLCP-TPU fabrics, and then dried in vacuum at 50 °C overnight. The average diameter of obtained CLCP-TPU fibers is ~1.5 μm (Fig. 1e). Subsequently, the CLCP-TPU fabrics were swollen in pure ILs for 12 h. Herein, the ILs are hydrophobic 1-ethyl-3-methylimidazolium bis-(trifluoromethylsulfonyl)imide ([EMIm][TFSI]), which possess high voltage stability, humidity resistance and strong self-adhesion stability[31]. After being dried in vacuum at room temperature for 6 h, the free-standing ILCP fabrics were finally obtained with the fibers swollen up to ~2.0 μm in diameter (Fig. 1f). The ILCP fabrics exhibit the mechanical properties of 240% elongation and 5.2 MPa tensile strength

(Supplementary Fig. 4), profiting from the elasticity of TPU and plasticization of ILs, fulfilling the needs of electronic skins[32].

**Performance of the ILCP-based device.** The IL content and ionic conductivity of ILCPs are affected by the immersion time (Fig. 2a). After immersion in ILs for 12 h, the IL content increased to 69% and the conductivity reached to 0.855 mS cm$^{-1}$, meeting the needs of electronics[33]. In general, flowable ILs within the polymer matrix are liable to exude under heating or vacuum, seriously losing the performance of devices[34]. Figure 2b compares the changes of IL content of CLCP-TPU fabrics and non-crosslinked LCP-TPU fabrics with time. The samples were treated under vacuum at room temperature for 24 h and subsequently at 50 °C for another 24 h. It is clear that the IL content of CLCP-TPU fabrics slightly decreased from 76% to 64% in the initial 3 hours and then maintained at the platform. In contrast, for non-crosslinked LCP-TPU fabrics, the IL content was severely falling to 22% without a platform. Therefore, the crosslinked networks and PEI complexation effectively restrain the exudation of ILs. The response to UV illuminance of ILCP-based device is illustrated in Fig. 2c, d. The device exhibited an increased current signal in response to UV light (Supplementary Fig. 5). When the

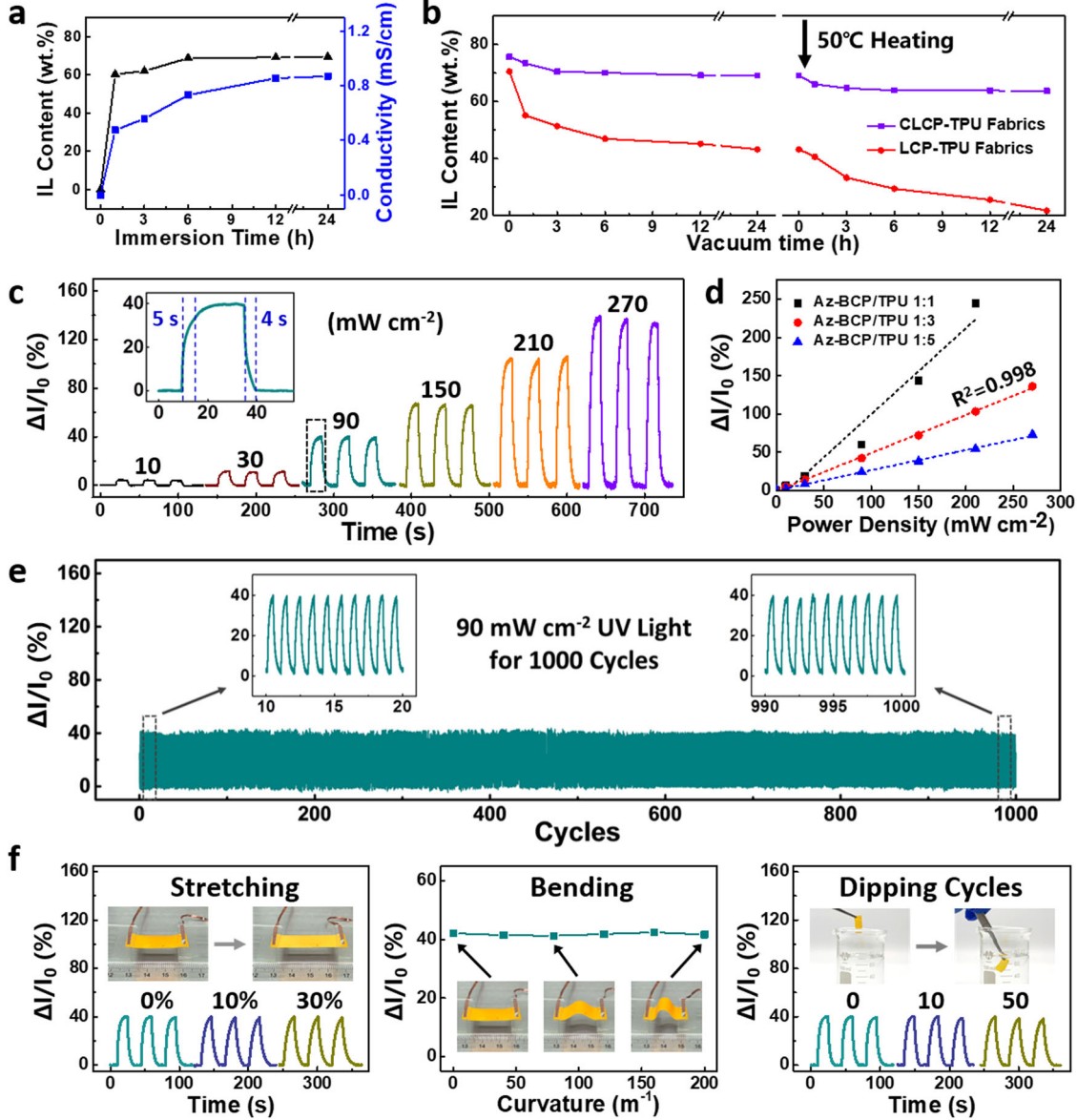

**Fig. 2 Performance of the ILCP-based device. a** Plots of IL content and ionic conductivity *versus* immersion time. Before the measurement of IL contents, samples were dried in vacuum at room temperature for 6 h. **b** Plots of IL content versus vacuum time of CLCP-TPU and LCP-TPU fabrics at room temperature and 50 °C. The measurement of IL contents was directly conducted after immersing samples in ILs for 12 hours. **c** Relative current changes in accordance with UV power density from 10 to 270 mW cm$^{-2}$. The relative current change ($\Delta I/I_0$) is defined as the ratio of current shift over the initial current, where $\Delta I$ and $I_0$ denote the measured current shift upon UV exposure and the initial current, respectively. The inset shows the response time of 5 s. The weight ratio of Azo-BCP/TPU in ILCPs is 1:3. **d** Experimental data and fitting curves of relative current changes *versus* UV power density in different Azo-BCP/TPU weight ratios. **e** Device response during consecutive 1000 UV on/off cyclic operation upon 90 mW cm$^{-2}$ UV light. The insets are the initial (left) and last (right) ten cycles of the test. **f** Relative current changes upon 90 mW cm$^{-2}$ UV light, when the ILCPs are subject to uniaxial stretching (0–30% strain), bending (0–200 m$^{-1}$ curvature), and repeated dipping in water (0–50 cycles), respectively. The device was fixed on a linear motor to control the strain and curvature.

weight ratio of Azo-BCP/TPU was 1:3, the fitted relationship between relative current changes and UV power density was calculated by linear regression method, as shown below:

$$\Delta I/I_0 = 0.50 \times P - 1.78 \qquad (1)$$

Where $\Delta I/I_0$ is the relative current change and $P$ represents UV power density. The slope of the fitting curve stands for the responsivity of ILCP-based device.

The device demonstrated an illuminance detection range of 10–270 mW cm$^{-2}$, a response time of 5 s, and a linear correlation coefficient of 0.998. When the weight ratio of Azo-BCP/TPU increased to 1:1, the linear detection range reduced because of

macroscopical bending deformation of ILCPs under strong UV illuminance (Supplementary Fig. 6). The phenomenon can be explained by the competitive relation between stress and optical path length in the thickness direction of films[26–28]. When the weight ratio of Azo-BCP/TPU is 1:1, CLCPs generate large contraction stress, resulting in significant current changes. However, the UV light only arrives to the surface region of CLCP films due to the large molar extinction coefficient of azobenzene, leading to the heterogeneous structure and then macroscopical bending deformation. When the weight ratio of Azo-BCP/TPU decreases to 1:5, the UV light can arrive to the deeper region of CLCP films, resulting in microscopic

deformation, and detectable current changes. The responsivity of ILCP-based device reduced with the decrease of Azo-BCP content. Upon 90 mW cm$^{-2}$ UV light, the relative current change ($\Delta I/I_0$) was constantly stable in static testing, undergoing 1000 testing cycles, 30% strain stretching and 200 m$^{-1}$ curvature bending, due to the intrinsic flexibility of ILCPs (Fig. 2e, f). Especially, the device can sustain 50 dipping cycles in water because of the great humidity resistance of ILs. The performance almost remained unchanged after 30 days at room temperature, and maintained stable upon dynamic 1000 cycles of stretching and folding operation (Supplementary Fig. 7). Compared with other reported wearable photodetectors, the comprehensive performance (such as flexibility and response time) of the ILCP-based device is excellent (Supplementary Table 1). Furthermore, the device exhibited a decreased current signal during dynamic stretching and bending, because of the increase in resistance (Supplementary Fig. 8). The negative current shift can be facilely distinguished from the positive ones upon UV response, extending the practical applications of the device.

**Working mechanism of the ILCP-based device.** To further investigate the working mechanism of the ILCP-based device, the optical microscopic (OM), polarizing optical microscopic (POM) and mechanical testing were conducted to illustrate the in-situ characterizations of ILCP fibers upon UV light (Fig. 3a–c and Supplementary Figs. 9 and 10). The POM images reflect the orientation of LC alignments through electrostatic fields of electrospinning. Because of contraction of the single fiber upon UV light, the fiber networks deform microscopically and generate contraction stress macroscopically. The alignment of mesogens was changed upon UV irradiation (Supplementary Movie 2).

When turning off UV light, the contraction stress is immediately released due to the excellent elasticity of TPU. To analyze the effect of elasticity, we compared the response and recovery time of CLCP-TPU fabrics with those of low-elastic CLCP-poly-acrylonitrile (PAN) fabrics (Supplementary Fig. 11a, b). The response and recovery time of CLCP-TPU fabrics are 5 s and 4 s, respectively, much faster than those of CLCP-PAN fabrics (11 s and 22 s, respectively). The crosslinked networks also facilitate the fast response of the device (Supplementary Fig. 11c). It is reasonable that the elastic TPU matrix can effectively transfer the UV light-triggered contraction stress to ILs and then accelerate ions migrating, converting to electric signals rapidly. On the other hand, it is reported that the mechanical stretch tends to align the LC mesogens along the stretching direction, referred as mechano-alignment[35]. As shown in the UV-Vis absorption spectra of the ILCP fabrics before and after UV irradiation (Supplementary Fig. 12), the peak intensity at 365 nm is corresponding to the $\pi$-$\pi^*$ transition of the *trans*-azobenzene. The *trans*-Azo content is defined as the ratio of absorption intensity over the pristine absorption intensity at 365 nm. The *trans*-Azo content decreased fast at the initial stage, and then levelled off to a plateau of 98.8 mol% at 10 s. Afterward, the *trans*-Azo content recovered to 99.6 mol% upon turning off UV irradiation for 60 s. The absorption intensity decreased/recovered completely in the subsequent cycles upon alternative UV on/off operation. It can be seen that the *cis*-azobenzene in fabrics can transfer to *trans*-azobenzene spontaneously upon turning off the UV light. The difference of *trans*-Azo content in one cycle is ~0.8 mol%, consistent with the reported values[26]. Thus, it is reasonable that the recovery stress of TPU can dominate the mechano-alignment of CLCPs, endowing a fast cyclic operation of the device in the absence of visible light or heating. The combination of CLCPs, ILs and TPU constructs an

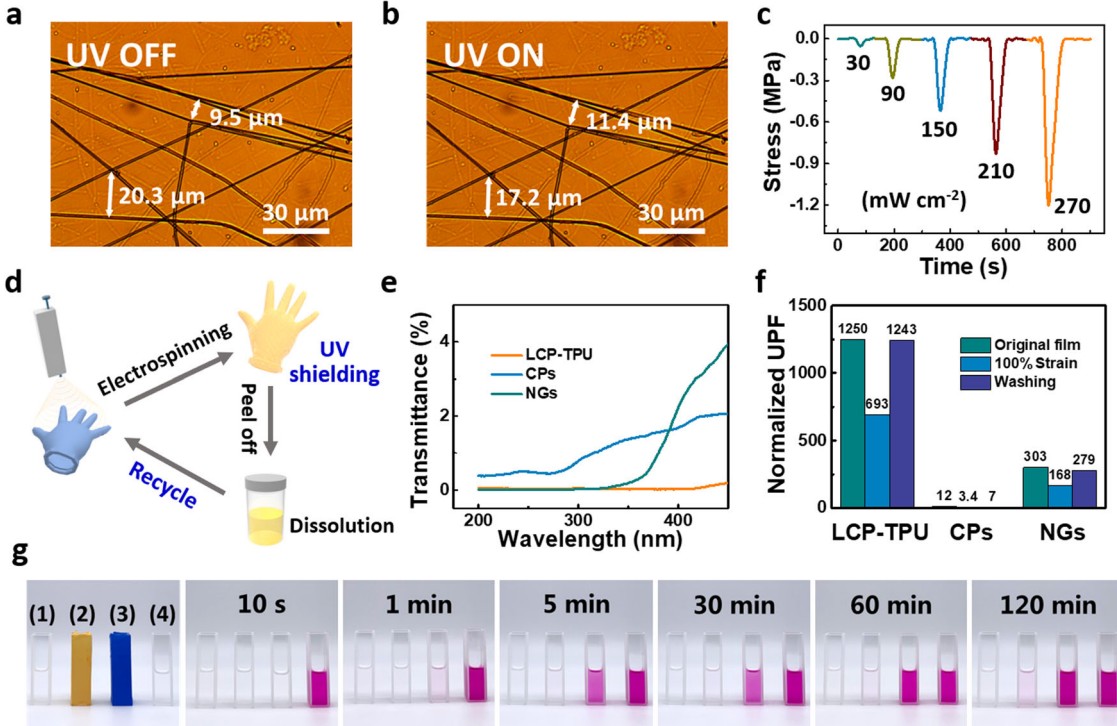

**Fig. 3 Working mechanism and UV shielding ability.** OM images of the ILCP fiber networks **a** before and **b** after UV irradiation. **c** UV induced contraction stress curves in accordance with UV power density. **d** Schematic diagram of the conformal and recyclable UV shielding LCP fabrics. **e** Transmittance spectra of the LCP-TPU fabrics, commercial products (CPs), and nitrile gloves (NGs), respectively. **f** The normalized UPF of original samples, 100% strain stretched samples and water washed samples, respectively. The normalized UPF is defined as the ratio of UPF over the film thickness. **g** Photographs of the UV shielding property evaluation upon 20 mW cm$^{-2}$ UV light by photochromic spiropyran indicator (10$^{-5}$ mol/L, ethanol solution). The four samples are (1) blank group, (2) LCP shielding group, (3) nitrile shielding group, and (4) no shielding group, respectively.

effective conversion mechanism of UV illuminance–mechanical stress–electric signals, bringing in rapid response time for ILCP-based devices.

**UV shielding ability of composite fabrics**. The LCP-TPU, CLCP-TPU and ILCP fabrics have a similar UV shielding ability, because they have almost the same UV transmittance spectrum. To simplify the fabrication process, the as-electrospun LCP-TPU fabrics can be directly utilized as recyclable UV shielding materials (Fig. 3d). A handheld electrospinning device is used to prepare conformal UV shielding fabrics facilely (Supplementary Fig. 13)[36]. Moreover, the LCP-TPU fabrics can redissolve in DMF/THF solutions for the recycling fabrication. To evaluate the UV shielding property, the ultraviolet protection factor (UPF) was calculated based on the GB/T 18830-2009 evaluation method, detailed in Supplementary Tables 2–4. When UPF > 50 and UV-A light transmittance < 5%, the sample can be identified as UV protective textiles. The calculated UPF of LCP-TPU fabrics (20 μm thickness) was initially 25040. Considering of the practical working conditions, the UPF decreased to 13851 at 100% strain and was almost unchanged upon washing. The UV shielding property and stability of LCP-TPU fabrics is much better than those of commercial products of UV protection clothing, nitrile gloves and previously reported UV shielding materials (Fig. 3e, f)[11,37]. However, the calculation parameters of UPF are based on solar UV irradiation (up to ~ 3 mW cm$^{-2}$) in daily life. Under some conditions such as lithography and photocuring, the UV illumination can reach to or higher than 20 mW cm$^{-2}$. Thus, spiropyran was selected as a sensitive photochromic indicator to evaluate the UV shielding property of LCP-TPU fabrics via spectrophotometry (Fig. 3g and Supplementary Fig. 14). Upon 20 mW cm$^{-2}$ UV light, the LCP-TPU fabrics exhibited the long-time shielding ability of 120 min, while the nitrile gloves lost efficacy within 1 min. Furthermore, the LCP-TPU fabrics can be reused after irradiation with 520 nm visible light (200 mW cm$^{-2}$) for 10 min. For practical applications, the LCP-TPU fabrics can be applied to prevent the UV ageing of nitrile gloves or skins (Supplementary Fig. 15).

**Applications of UV security encoding**. As discussed above, the electric signal output of the ILCP-based device allows it for the facile integration into the Internet of Things system. The ILCP-based device can not only record the real-time UV illuminance on liquid crystal display (LCD) screen, but also transfer data to smartphones through Bluetooth for remote monitoring (Supplementary Figs. 16 and 17 and Movie 3). Furthermore, UV light can avoid being intercepted and interfered because of its short wavelength, which can be used for security communication. Under various extreme environments in the military field, it is highly desired for a portable and robust UV detector to guarantee communication quality. Thus, we have designed wearable information encoding and encryption applications based on ILCP fabrics, such as UV unlock and UV input. As illustrated in Fig. 4a, the UV coding system is consisted of an ILCP-based encoding part and an Arduino singlechip-based decoding part.

The physical connection diagram and encoding logic of UV security unlock are displayed in Fig. 4b, c. If the ILCP array composed of three ILCP-based devices is irradiated with the UV illuminance of "20, 30, 40" mW cm$^{-2}$ in right sequence, the green LED will be turned on, indicating a unlock command. Otherwise, the red LED will be turned on, indicating a lock command (Fig. 4f and Supplementary Movie 4). We also conducted the application of UV security input (Fig. 4d, e). When the alphabets of A~Z are critically corresponded to the determined UV illuminance, the UV illuminance information can be recognized and read out by a smartphone, resulting in the input of "BUAA" on the screen

(Fig. 4g and Supplementary Movie 5). Moreover, the information can be reprogrammed and transformed by simply adjusting the code. The present work opens up a paradigm for UV security information storage and communication based on wearable and robust UV responsive moiety-containing polymer detectors.

In summary, we demonstrate a robust ILCP-based UV monitoring and shielding device via the combination of electrospun CLCP-TPU fabrics and hydrophobic ILs, where CLCPs work as sensor components, ILs offer the electric signal output, and TPU contributes to high elasticity and fast response. The signal conversion, that is UV illuminance–mechanical stress–electric signals, significantly improves the sensitivity of flexible UV detectors, benefiting from the cooperative effects of Azo-LC alignment and polymer networks. The excellent elasticity of TPU facilitates the fast cyclic operation of the device without visible light or heating. The device exhibits a wide UV illuminance detection range of 10 ~ 270 mW cm$^{-2}$, a response time of 5 s and a recovery time of 4 s. In addition, the abundant amino groups in PEI and urethane groups in TPU make the device easily adhere to skins by multiple hydrogen bonds. The strong electrostatic interactions between amino groups and ions effectively restrain the exudation of ILs. Thus, the device maintains stable performance upon 1000 UV on/off testing cycles, 30% strain stretching, 200 m$^{-1}$ curvature bending, and especially 50 dipping cycles in water. Furthermore, the as-electrospun recyclable LCP-TPU fabrics have the long-time shielding ability of 120 min upon 20 mW cm$^{-2}$ UV light, with a high UPF of 25040. The LCP fabrics can be reused after irradiation with 520 nm visible light (200 mW cm$^{-2}$) for 10 min, applying to prevent the UV ageing of skins or other materials. For practical applications in various extreme environments, demonstrations of information encoding and encryption based on wearable ILCPs are successfully realized for UV unlock and UV input via the Internet of Things technologies, benefiting from the portability and robustness of ILCPs. The performance of ILCPs is expected to be further enhanced through the improvement of the device structure and testing circuit. The well-defined composites of ILCPs and signal conversion mechanism provide brand-new opportunities for functional polymer-based sensors to fabricate various sensitive wearable electronics.

## Methods
**Materials**. Tetrahydrofuran (THF, 99.5%, Aladdin), N, N-dimethylformamide (DMF, 99.9%, Aladdin), Ethanol (EtOH, 99.8%, Aladdin), Branched poly-ethyleneimine (PEI, $M_w$=1×10$^4$ g/mol, 99%, Aladdin), Thermoplastic polyurethane (TPU, 1190 A, Elastollan), and 1-ethyl-3-methylimidazolium bis-(tri-fluoromethylsulfonyl)imide ([EMIm][TFSI], 99%, Adamas) were used as received. The block copolymers (PEO$_{45}$-$b$-PMA(Az)$_{28}$, $M_n$=1.82×10$^4$ g/mol, $M_w$/$M_n$=1.13), were synthesized by atom transfer radical polymerization (ATRP), detailed in Supplementary Fig. 1[29,38,39].

**Fabrication of the ILCP-based device**. Firstly, Azo-BCPs (80 mg) and TPU (240 mg) were dissolved in DMF and THF mixture (1 mL, volume ratio 1:1) via stirring for 12 h at room temperature. A self-assembly electrospinning device was used to produce fibers with a positive DC voltage of 15 kV and a feed rate of 1 mL/h. The distance between the extrusion nozzle and the roller was set at 15 cm. The LCP-TPU fabrics of 20 μm thickness were collected on aluminum foil wrapped around a rotating roller. The LCP-TPU fabrics were immersed in an ethanol solution of PEI (2 mg/mL) for 6 h and then dried in vacuum overnight. Subsequently, the obtained CLCP-TPU fabrics were swollen in pure ILs for 12 h. The fabrics were sandwiched in filter papers, clamped by two glass sheets, and then dried in vacuum (0.02 Mpa) at room temperature for 6 h. The free-standing ILCP fabrics were finally obtained. For the UV response testing of ILCP fabrics (10 × 25 mm$^2$), copper wires were attached at the opposite sides of the ILCP fabrics by silver paste.

**Characterization and measurements**. Scanning electron microscope (SEM) images were obtained on a TESCAN VEGA3 microscope. UV light at 365 nm was obtained from Omron (ZUV-C30H) LED irradiator. Visible light at 520 nm was obtained from CCS (HLV-22GR-3W) LED irradiator. The IL content is defined as the ratio of ($w$-$w_0$)/$w$, where $w_0$ and $w$ denote the measured weights of CLCPs

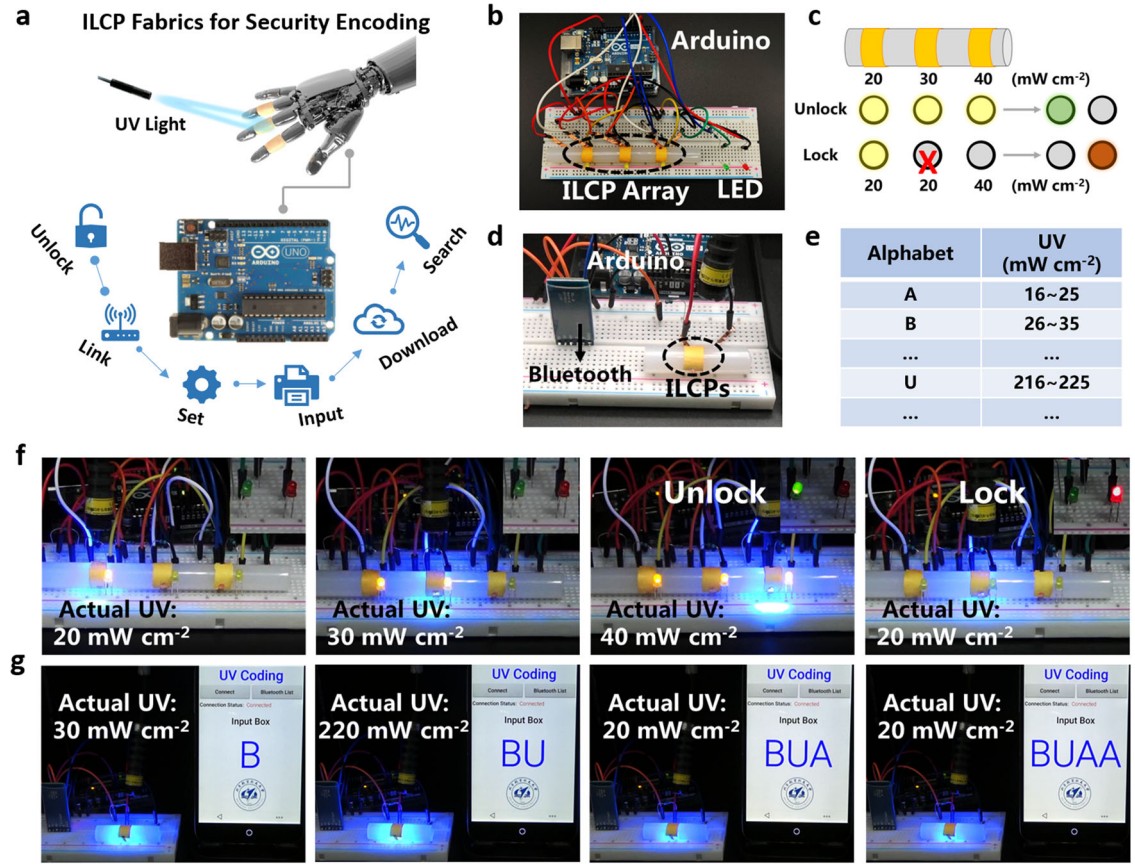

**Fig. 4 Applications of UV security encoding. a** Schematic illustration of the practical application of ILCP fabrics for security encoding. The decoding of Arduino singlechip can be applied to unlock, link, set, input, download, search, and other communication forms. **b** The physical connection diagram and **c** encoding logic of UV security unlock. The right UV illuminance series is "20, 30, 40" mW cm⁻². **d** The physical connection diagram and **e** encoding logic of UV security input. Photographs of the practical working process of **f** UV unlock and **g** UV input.

before and after immersion in ILs, respectively. The weights of CLCPs were measured by an analytical balance (METTLER TOLEDO ME104E). The current response was monitored by an electrometer (Keithley 6517B) with an applied voltage of 0.1 V. In the dipping experiment, the sample was dipped in water immediately between cycles. Before performance testing, the sample was suffi-ciently dried at room temperature for 6 h. The tensile properties were measured in a stretch mode at a strain rate of 3 mm/min at room temperature on Xieqiang CTM2050 mechanical analyser. The dynamic mechanical analysis was carried out by a mechanical analyzer (TA Instruments, Waters Ltd., DMA Q800) in stretching mode. The temperature was set from −70 °C to 100 °C, with a heating rate of 5 °C/min. Optical microscopic (OM) and polarizing optical microscopic (POM) experiments were conducted by Shang Guang 59XF microscope. The recyclable UV shielding fabrics were prepared by a handheld electrospinning device (Junada MPEG-1). The UV shielding properties were evaluated by UV-vis transmittance spectra (Shimadzu UV-2600 spectrophotometer). The commercial UV protection clothing was purchased from Decathlon (90 μm). The Android application of UV encoding was made by MIT App Inventor 2. All the photos and movies were recorded on a SONY HDR-CX450 digital video camera.

## Data availability

The data that support the findings of this study are available from the corresponding author upon reasonable request.

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

## Acknowledgements

This work was financially supported by the National Natural Science Foundation of China (No. 51472018), the Natural Science Foundation of Beijing Municipality (No. 2202024), and the Fundamental Research Funds for the Central Universities.

## Author contributions

X.Z. and A.C. identified the initial research directions. X.Z. and Y.J. performed the experiments and analyzed the data. All authors contributed to the writing of manuscript.

## Competing interests

The authors declare no competing interests.
