## [Peer Review File · Nature Communications]

REVIEWER COMMENTS

Reviewer #1 (Remarks to the Author):

In this work, the authors have introduced well-defined polymer composite fabrics consisted of Azo-CLCPs, ILs and TPU to realize a robust UV detector with significantly accelerated photo-response speed and sensitivity. A novel and effective UV illuminance-mechanical stress-electric signal conversion was proposed to improve photoelectric performance. The exudation of ILs was restrained by the strong electrostatic interactions between crosslinked networks and ions, endowing good stability of the detectors. They further designed the wearable ILCP-based on-demand information encoding electronics for UV security unlock and input, benefiting from electrospinnability and good performance of ILCPs. The manuscript is well written, and the concept is worthy of investigation. Intrinsically flexible polymer sensors with excellent photoelectric performance are highly needed in the field of wearable devices. In light of the above-mentioned points, I suggest that this manuscript can be accepted to Nature Communications after the following minor revision.

1. To further express the robustness of intrinsically flexible polymer UV detectors, the device performance after consecutive stretching or bending cyclic operation should be investigated.
2. In Fig. 2f, has the sample been sufficiently dried after dipping in water? The dipping time between cycles should also be indicated.
3. It would be better if the device of non-crosslinked LCP-TPU fabrics is added to the performance comparison data.

Reviewer #2 (Remarks to the Author):

In this work, the authors presented an azobenzene-based UV sensor having high flexibility, fast response, and linearity toward the intensity of UV irradiation. Even though the authors showed many interesting application results such as mechanical tests, water-proofness, and feasible security encodings, it is hard to catch and find what novelties this work has. The key material (PEI-derivative azobenzene-based polymer) has been already reported in authors' previous work (Small, 2019) and its liquid crystalline form has been also presented elsewhere (Polymer Chemistry, 2021). In this work, the authors have used TPU for elastic backbone, but such combination has been well known for developing extensible devices. Therefore, the current manuscript is hard to be accepted in Nat. Comm. and it is mandatory that the manuscript should be fully revised with resolving following issues.

- 1) In terms of novelty, the authors must address what material system is original and why it has strong impact on the suggesting UV-sensing device.
- 2) The authors must address that the Azo-BCP(PEI)/TPU system is stable in terms of solubility and phase separation. Does the Azo-BCP(PEI) have enough solubility with TPU? In TPU matrix, how do the authors prevent the aggregation and crystallization of azobenzene? Why is the suggesting concentration of the Azo-BCP(PEI) optimal?
- 3) Even for the ratio 1:5 of Azo to TPU, UV-response was successfully measured in Fig. 2(d). What is the reason that such low fraction of Azo system can show high current changes? Can the authors show the relative UV-Vis measurement results for each different weight ratios?

4) The presented bending test was somewhat poorly carried out in terms of scientific way, because the bending angle shown in Fig. 2(f) is relatively measured on the rubbery glove. Provide bending radius or bending curvature (strain). Also, the current change must be measured at the circumstance where the whole part is under the same bending mode. Current demonstration shows mixed mode consisting of stationary and local bending together.

5) The resulting flexibility was measured on the textile substrate, so it is not the result from the pure Azo-CLCP. Is it possible to provide the values of only the Azo-CLCP without textile?

Reviewer #3 (Remarks to the Author):

The authors report on a novel ionic liquid containing azobenzene liquid crystalline polymer, polyurethane composite that can be used as an ultraviolet detector and shielding device. Upon UV illumination a contraction stress is generated in the composite that is subsequently converted into an electrical signal. The response time of the device is fast (5s). The performance of the devices is stable during many testing cycles and upon stretching, bending and washing. Such a device is novel and interesting from an application point of view.

However, the mechanism of the device is not clear. Furthermore, the morphology and interactions between the different component is not clear.

Main issues:

- 1) The authors mention the presence of electrostatic interactions and hydrogen bonds between ILs and LC networks. What is the direct experimental evidence for this statement?
- 2) The contraction stress is immediately released upon turning off the UV light. This suggests that photothermal defects dominate as the half lifetime of cis azobenzene is more than a day. The authors should determine the cis-trans azobenzene ratio and the temperature in the composite material before and after light exposure.
- 3) The POM images are confusing. Some fibers bend while other contract. Some fibers do not actuate at all. The use of cross polarizer might help to determine the alignment in the samples.
- 4) It would be interesting to know the performance of the device when illuminated with sunlight and indoor light. This might be important for daily applications.
- 5) LCP-TPU fabrics can be reused but what about the ILCP fabrics.

The list of changes in revised manuscript is shown below:

1. FTIR spectra of PEI, IL, and PEI/IL blends were added in Figure S2 to prove the presence of electrostatic interactions and hydrogen bonds between ILs and LC networks.
2. Dynamic mechanical analysis (DMA) was added in Figure S3a to investigate the compatibility of Azo-BCP/TPU. The section of “Fabrication of ILCP fabrics” was revised to illustrate the stable Azo-BCP/TPU system in terms of compatibility.
3. UV-Vis absorption spectra of Azo-BCP/TPU fabrics were added in Figure S3b to demonstrate 365 nm UV absorption in different weight ratios.
4. The device performances in 1:1 and 1:5 Azo-BCP/TPU weight ratios were added in Figure S6 to supplement the data in Figure 2d.
5. The device performance after consecutive stretching and folding cyclic operation was added in Figure S7 to further express the robustness of wearable UV detectors.
6. The bending test was improved in Figure 2f according to the reviewer’s suggestion.
7. The section of Methods was revised to illustrate the dipping experiment in detail.
8. The POM images of the texture of the ILCP fibers before and after UV irradiation were added in Figure S9 and Movie S2 to further prove the microscopical deformation of ILCP fibers.
9. The device performance of non-crosslinked LCP-TPU fabrics was added in Figure S11 to illustrate the significance of crosslinked networks.
10. UV-Vis absorption spectra of the ILCP fabrics before and after UV irradiation were investigated in Figure S12 to determine the *trans-cis* azobenzene ratio. The section of “Working mechanism of the ILCP-based device” was revised to further explain the working mechanism.

The following is the reply to the reviewers.

To begin, we would like to appreciate the reviewers for their thoughtful and constructive comments. The reviewers’ remarks have been very carefully considered and we tried to clarify their concerns. Please see below for a point-by-point response to the reviewers’ specific comments.

Reviewer #1 (Remarks to the Author):

In this work, the authors have introduced well-defined polymer composite fabrics consisted of Azo-CLCPs, ILs and TPU to realize a robust UV detector with significantly accelerated photo-response speed and sensitivity. A novel and effective UV illuminance-mechanical stress-electric signal conversion was proposed to improve photoelectric performance. The exudation of ILs was restrained by the strong electrostatic interactions between crosslinked networks and ions, endowing good stability of the detectors. They further designed the wearable ILCP-based on-demand information encoding electronics for UV security unlock and input, benefiting from electrospinnability and good performance of ILCPs. The manuscript is well written, and the concept is worthy of investigation. Intrinsically flexible polymer sensors with excellent photoelectric performance are highly needed in the field of wearable devices. In light of the above-mentioned points, I suggest that this manuscript can be accepted to Nature Communications after the following minor revision.

Reply:

We sincerely thank the reviewer for careful review and generous comments on our work. We hope our revised manuscript could match the requirement of *Nature Communications* with an improved quality.

1. To further express the robustness of intrinsically flexible polymer UV detectors, the device performance after consecutive stretching or bending cyclic operation should be investigated.

Reply:

We highly appreciate the reviewer for this suggestion. The device performance after consecutive stretching and bending cyclic operation was investigated in Supplementary Information (Figure S7), as shown in Figure R1. To further express the robustness and flexibility of the polymer UV detectors, the device was subject to 30% strain stretching and 180° bending (folding), assisted by a tweezer. The performance almost remained unchanged after 1000 stretching and folding cycles.

Figure R1. (a) Schematic diagram of the consecutive stretching and folding cyclic operation. A 30% strain stretching operation plus a folding operation denotes as one cycle. The device was fixed on a linear motor to control the cyclic operation. A tweezer was used to fold the device. (b) The performance of ILCP-based device during consecutive 1000 stretching and folding cycles. The relative current changes were recorded upon 90 mW cm^{-2} UV light.

2. In Fig. 2f, has the sample been sufficiently dried after dipping in water? The dipping time between cycles should also be indicated.

Reply:

We appreciate the reviewer for the detailed evaluation and valuable comments. In the dipping experiment, the sample was dipped in water immediately between cycles. Before performance testing, the sample was sufficiently dried at room temperature for 6 h. The section of Methods was revised to illustrate the dipping experiment in detail.

3. It would be better if the device of non-crosslinked LCP-TPU fabrics is added to the performance comparison data.

Reply:

We highly appreciate the reviewer for this suggestion. The device performance of non-crosslinked LCP-TPU fabrics was added in Supplementary Information (Figure S11), as shown in Figure R2. It is suggesting that the crosslinked networks are essential for the fast response of the device.

Figure R2. The response and recovery time of non-crosslinked LCP-TPU fabrics upon 90 mW cm⁻² UV light.

Reviewer #2 (Remarks to the Author):

In this work, the authors presented an azobenzene-based UV sensor having high flexibility, fast response, and linearity toward the intensity of UV irradiation. Even though the authors showed many interesting application results such as mechanical tests, water-proofness, and feasible security encodings, it is hard to catch and find what novelties this work has. The key material (PEI-derivative azobenzene-based polymer) has been already reported in authors' previous work (Small, 2019) and its liquid crystalline form has been also presented elsewhere (Polymer Chemistry, 2021). In this work, the authors have used TPU for elastic backbone, but such combination has been well known for developing extensible devices. Therefore, the current manuscript is hard to be accepted in Nat. Comm. and it is mandatory that the manuscript should be fully revised with resolving following issues.

Reply:

We are very grateful for the reviewer's careful review and constructive comments on our work in detail. The novelties in this work have been illustrated in the following replies. We hope our revised manuscript could match the requirement of *Nature Communications* with an improved quality.

1) In terms of novelty, the authors must address what material system is original and why it has strong impact on the suggesting UV-sensing device.

Reply:

We appreciate the reviewer for the detailed evaluation and valuable comments. In this

work, we demonstrate well-defined **composite fabrics consisted of Azo-CLCPs, ionic liquids (ILs), and thermoplastic polyurethane (TPU)** for intrinsically flexible UV detectors *for the first time*, which is a *unique and innovative material system*. Upon UV irradiation, Azo-CLCPs deform *microscopically* and generate contraction stress. The composite fabrics constructs an effective conversion mechanism of UV illuminance-mechanical stress-electric signals, where CLCPs work as sensor components, ILs offer the electric signal output, and TPU contributes to high elasticity and fast response. The *functions* demonstrated by this material system are novel.

Generally, CLCPs are highly sensitive to environmental stimuli due to the synergistic effect of LC alignment and polymer networks, generating reversible *macroscopical* shape changes, which are mainly utilized in the field of soft actuators (for example, Yanlei Yu*. *Adv. Mater.* **2019**, 31, 1904224). To the best of our knowledge, **CLCPs used as wearable UV detectors or other flexible electronics have not been reported yet.**

From the standpoint of UV-sensing properties, the comprehensive performance (such as durability and response time) of the device is excellent compared with other reported wearable photodetectors. **Benefitting from the intrinsic flexibility and robustness of the composites**, the device can output electric signals and maintain stable performance upon stretching, bending (even for folding), washing, etc., which has significant potential applications in various extreme environments in the military field, such as UV communication.

It is an interdisciplinary research in the field of advanced functional materials, wearable electronics, and information security. **All innovations come from the well-defined novel composites.** We believe that it could show enough novelty to appeal to the broad readership of *Nature Communications*. The sections of Introduction and Results were revised to further address the novelty and originality of this work.

2) *The authors must address that the Azo-BCP(PEI)/TPU system is stable in terms of solubility and phase separation. Does the Azo-BCP(PEI) have enough solubility with TPU? In TPU matrix, how do the authors prevent the aggregation and crystallization of azobenzene? Why is the suggesting concentration of the Azo-BCP(PEI) optimal?*

Reply:

We highly appreciate the reviewer for this suggestion. In this work, CLCP-TPU

fabrics were fabricated by a two-step post-crosslinking method, which involved the molding of non-crosslinked Azo-BCP/TPU and the subsequent crosslinking in PEI solution. Thus, it is only necessary to investigate the solubility between Azo-BCPs and TPU. The used amphiphathic Azo-BCPs contain 11 wt.% hydrophilic PEO blocks, which have enough compatibility with TPU in principle. In addition, as shown in the photographs and SEM images (Figure 1d-f), there is no phase separation in the composite materials.

To further confirm it, dynamic mechanical analysis (DMA) was conducted to demonstrate the glass transition temperature (T_g) of Azo-BCP/TPU fabrics, as shown in Figure S3a (Figure R3). The two peaks of T_g were fused in each weight ratios, which confirmed the good solubility of Azo-BCP/TPU. The optimal concentration of the Azo-BCPs was determined by the subsequent device performance testing. The section of “Fabrication of ILCP fabrics” was revised to illustrate the compatibility of Azo-BCP/TPU.

Figure R3. Plots of $\text{tg } \delta$ versus temperature of Azo-BCP/TPU fabrics in different weight ratios.

3) Even for the ratio 1:5 of Azo to TPU, UV-response was successfully measured in Fig. 2(d). What is the reason that such low fraction of Azo system can show high current changes? Can the authors show the relative UV-Vis measurement results for each different weight ratios?

Reply:

We thank the reviewer for raising this question. The UV-Vis absorption spectra of Azo-BCP/TPU fabrics were added in Figure S3b (Figure R4). The fabrics exhibit 365 nm UV absorption in each weight ratios. The device performances of Azo-BCP/TPU fabrics in 1:1 and 1:5 weight ratios were added in Figure S6 (Figure R5). Even for the

low fraction of Azo-BCPs, the device can show considerable current changes.

The phenomenon can be explained by the competitive relation between stress and optical path length in the thickness direction of films (Yanlei Yu*. *Adv. Mater.* **2019**, 31, 1904224). It is reported that based on reasonably designed structures, as long as 1 mol% of azobenzene mesogens reach to the photostationary state upon illumination, the generated energy can lead to the phase transition of the whole systems, featuring a fast response time of several seconds for Azo-CLCPs (Tomiki Ikeda*. *Angew. Chem. Int. Ed.* **2007**, 46, 506). When the weight ratio of Azo-BCP/TPU increases to 1:1, CLCPs generate large contraction stress, resulting in significant current changes. However, the UV light only arrives to the surface region of CLCP films due to the large molar extinction coefficient of azobenzene, leading to the heterogeneous structure and then macroscopical bending deformation (Figure R6). When the weight ratio of Azo-BCP/TPU decreases to 1:5, the UV light can arrive to the deeper region of CLCP films, resulting in microscopic deformation, and detectable current changes. The section of “Performance of the ILCP-based device” was revised to explain the competitive relation between stress and optical path length.

Figure R4. UV-Vis absorption spectra of Azo-BCP/TPU fabrics in different weight ratios.

Figure R5. Relative current changes in accordance with UV power density from 10 to 270 mW cm^{-2} . The weight ratios of Azo-BCP/TPU in ILCPs are (a) 1:1 and (b) 1:5.

Figure R6. Schematic diagram of the competitive relation between stress and optical path length in the thickness direction of films.

4) The presented bending test was somewhat poorly carried out in terms of scientific way, because the bending angle shown in Fig. 2(f) is relatively measured on the rubbery glove. Provide bending radius or bending curvature (strain). Also, the current change must be measured at the circumference where the whole part is under the same bending mode. Current demonstration shows mixed mode consisting of stationary and local bending together.

Reply:

We sincerely appreciate the reviewer for this suggestion. The bending test was improved in Figure 2f (Figure R7). The device was fixed on a linear motor to control the bending curvature. The device performance maintains stable upon bending of 0 ~ 200 cm^{-1} curvature.

Figure R7. Relative current changes upon bending of 0 ~ 200 cm^{-1} curvature. The data were recorded upon 90 mW cm^{-2} UV light.

5) The resulting flexibility was measured on the textile substrate, so it is not the result from the pure Azo-CLCP. Is it possible to provide the values of only the Azo-CLCP without textile?

Reply:

We appreciate the reviewer's valuable comments. The mechanical properties of CLCP casting films were investigated in Figure R8. Previous Figure S4 contained the stress-strain curves of CLCP-TPU fabrics and ILCP (CLCP-TPU-IL) fabrics. The CLCPs without textile exhibit the mechanical properties of 22% elongation and 1.7 MPa tensile strength, which have greater flexibility than rigid semiconductors. Benefiting from the good compatibility of Azo-BCP/TPU, the CLCPs in fabrics sustain much larger elongation.

Figure R8. The stress-strain curves of the CLCP-TPU fabrics, ILCP fabrics and CLCP casting films.

Reviewer #3 (Remarks to the Author):

The authors report on a novel ionic liquid containing azobenzene liquid crystalline polymer; polyurethane composite that can be used as an ultraviolet detector and shielding device. Upon UV illumination a contraction stress is generated in the composite that is subsequently converted into an electrical signal. The response time of the device is fast (5s). The performance of the devices is stable during many testing cycles and upon stretching, bending and washing. Such a device is novel and interesting from an application point of view.

However, the mechanism of the device is not clear. Furthermore, the morphology and interactions between the different component is not clear.

Reply:

We sincerely thank the reviewer for careful review and generous comments on our work. We hope our revised manuscript could match the requirement of *Nature Communications* with an improved quality.

Main issues:

1) The authors mention the presence of electrostatic interactions and hydrogen bonds between ILs and LC networks. What is the direct experimental evidence for this statement?

Reply:

We sincerely appreciate the reviewer for this suggestion. The electrostatic interactions and hydrogen bonds between ILs and LC networks mainly come from the interactions between ILs and PEI. To simplify the analysis of peak shifting, FTIR spectra of PEI, IL, and PEI/IL blends were investigated in Figure S2 (Figure R9). The shifting of N-H stretching bands from PEI, C_{4,5}-H stretching bands from cations, and C-F stretching bands from anions can prove the presence of electrostatic interactions and hydrogen bonds between ILs and PEI. Thus, it is confirmed that there are electrostatic interactions and hydrogen bonds in ILCPs. In addition, the changes of IL content upon heating and vacuum in Figure 2b also demonstrated that the crosslinked networks and PEI complexation can effectively restrain the exudation of ILs.

Figure R9. FTIR spectra of PEI, IL, and PEI/IL blends between 900 ~ 3500 cm^{-1} . The weight ratio of PEI/IL blends was 1:1.

2) The contraction stress is immediately released upon turning off the UV light. This suggests that photothermal defects dominate as the half lifetime of *cis* azobenzene is more than a day. The authors should determine the *cis-trans* azobenzene ratio and the temperature in the composite material before and after light exposure.

Reply:

We appreciate the reviewer for the professional review on our article. The *cis-trans* azobenzene ratio was determined by the UV-Vis absorption spectra of the ILCP fabrics before and after UV irradiation, as shown in Figure S12 (Figure R10). Because the absorption intensity of 20 μm thick ILCP fabrics is over the detection range of the spectrophotometer, the thickness of studied ILCP fabrics is 5 μm . The peak intensity at 365 nm is corresponding to the $\pi-\pi^*$ transition of the *trans*-azobenzene. The *trans*-Azo content is defined as the ratio of absorption intensity over the pristine absorption intensity at 365 nm. The *trans*-Azo content decreased fast at the initial stage, and then levelled off to a plateau of 98.8 mol% at 10 s. Afterward, the *trans*-Azo content recovered to 99.6 mol% upon turning off UV irradiation for 60 s. The absorption intensity decreased/recovered completely in the subsequent cycles upon alternative UV on/off operation. It can be seen that the *cis*-azobenzene in fabrics can transfer to *trans*-azobenzene spontaneously upon turning off the UV light. The difference of *trans*-azobenzene in one cycle is 0.8 mol%, consistent with the reported values (Tomiki Ikeda*. *Angew. Chem. Int. Ed.* **2007**, 46, 506).

The temperature on ILCP fabrics and printing common papers (as blank experiment) were investigated by a probe thermometer, as shown in Figure R11. In either case, the

temperature slightly increased from 30.2 °C to 31.3 °C, coming from the UV-LED light source. In addition, the glass transition temperature of Azo-BCPs is ~ 70 °C. Therefore, the effect from UV-LED light source on *cis-trans* transition is negligible.

Thus, it is reasonable that the recovery stress of TPU can dominate the mechano-alignment of CLCPs, endowing a fast cyclic operation of the device in the absence of visible light or heating. The section of “Working mechanism of the ILCP-based device” was revised to further explain the working mechanism.

Figure R10. UV-Vis absorption spectra and plots of *trans*-Azo content versus time in the ILCP fabrics upon (a, b) turning on UV irradiation (10 mW/cm⁻²) and (c, d) turning off UV irradiation, respectively. The *trans*-Azo content is defined as the ratio of absorption intensity over the pristine absorption intensity at 365 nm. (e) UV-Vis absorption spectra of 3 irradiation cycles. A cycle is consisted of 10 s of turning on UV irradiation and 60 s of turning off UV irradiation. Film thickness: 5 μm.

Figure R11. The temperature on (a) ILCP fabrics and (b) printing common papers (as blank experiment) before and after UV light exposure.

3) The POM images are confusing. Some fibers bend while other contract. Some fibers do not actuate at all. The use of cross polarizer might help to determine the alignment in the samples.

Reply:

We greatly appreciate the reviewer's suggestion. The POM images of the texture of the ILCP fibers before and after UV irradiation were added in Figure S9 (Figure R12) and Movie S2. The alignment of mesogens was changed upon UV irradiation, which further proves the microscopical deformation of ILCP fibers.

Figure R12. POM images of the texture of the ILCP fibers before and after UV irradiation.

4) It would be interesting to know the performance of the device when illuminated with sunlight and indoor light. This might be important for daily applications.

Reply:

We appreciate the reviewer for this suggestion. The UV irradiation in sunlight is up to $\sim 3 \text{ mW cm}^{-2}$ (Outdoor, Summer noon in Beijing, China). However, the resolution of the device is about 5 mW cm^{-2} , which is hard to measure the UV irradiation in sunlight. The UV irradiation in indoor light (incandescent lamp, and avoid sunlight) is close to 0 mW cm^{-2} . All the performance testing of the device can be conducted upon

the indoor light. There is usually a trade-off between robustness and sensitivity in wearable electronics. In this work, we have demonstrated the material design, working mechanism and proper applications of the robust polymer UV detectors. The device performance is expected to be further enhanced through the improvement of the device structure and testing circuit at the next stage.

5) LCP-TPU fabrics can be reused but what about the ILCP fabrics.

Reply:

We thank the reviewer for raising this question. In this work, the LCP-TPU fabrics can be utilized as recyclable and reshapable UV shielding clothes. Then the ILCP-based devices adhere to the UV shielding fabrics for real-time monitoring of UV illuminance. The devices can be reused for long time benefitting from the robustness of the composite fabrics (Figure 2e, f and Figure S7). However, the ILCP fabrics cannot be recycled or reshaped after chemical crosslinking in PEI solution.

REVIEWERS' COMMENTS

Reviewer #1 (Remarks to the Author):

The authors have addressed my concerns and the paper is now acceptable for publishing.

Reviewer #2 (Remarks to the Author):

All raised issues and questions have been well described in the revised manuscript and authors' responses. The current form of the manuscript is now acceptable to Nat. Comm.

Reviewer #3 (Remarks to the Author):

The revised manuscript is now suitable for publication. Most issues have been adequately addressed.

The following is the reply to the reviewers.

To begin, we would like to appreciate the reviewers for their thoughtful and constructive comments. The reviewers' remarks have been very carefully considered and we tried to clarify their concerns. Please see below for a point-by-point response to the reviewers' specific comments.

Reviewer #1 (Remarks to the Author):

The authors have addressed my concerns and the paper is now acceptable for publishing.

Reply:

We sincerely appreciate the reviewer for careful review and generous comments on our work. We hope our revised manuscript could match the requirement of *Nature Communications* with an improved quality.

Reviewer #2 (Remarks to the Author):

All raised issues and questions have been well described in the revised manuscript and authors' responses. The current form of the manuscript is now acceptable to Nat. Comm.

Reply:

We are very grateful for the reviewer's careful review and constructive comments on our work in detail. We hope our revised manuscript could match the requirement of *Nature Communications* with an improved quality.

Reviewer #3 (Remarks to the Author):

The revised manuscript is now suitable for publication. Most issues have been adequately addressed.

Reply:

We sincerely appreciate the reviewer for the detailed evaluation and valuable comments. We hope our revised manuscript could match the requirement of *Nature Communications* with an improved quality.